# LG-CAV: Train Any Concept Activation Vector with Language Guidance

**Qihan Huang[1], Jie Song[1, †], Mengqi Xue[2], Haofei Zhang[1],**
**Bingde Hu[1], Huiqiong Wang[3], Hao Jiang[4], Xingen Wang[1, 5], Mingli Song[1]**
[1] Zhejiang University, [2] Hangzhou City University
[3] Ningbo Innovation Center, Zhejiang University
[4] Alibaba Group, [5] Bangsheng Technology Co., Ltd.
`{qh.huang,sjie,haofeizhang,tonyhu,huiqiong_wang,newroot,brooksong}@zju.edu.cn`
`mqxue@zucc.edu.cn, aoshu.jh@alibaba-inc.com`

## Abstract

Concept activation vector (CAV) has attracted broad research interest in explainable AI, by elegantly attributing model predictions to specific concepts. However, the training of CAV often necessitates a large number of high-quality images, which are expensive to curate and thus limited to a predefined set of concepts. To address this issue, we propose **L**anguage-**G**uided CAV (LG-CAV) to harness the abundant concept knowledge within the certain pre-trained vision-language models (*e.g.*, CLIP). This method allows training *any* CAV without labeled data, by utilizing the corresponding concept descriptions as guidance. To bridge the gap between vision-language model and the target model, we calculate the activation values of concept descriptions on a common pool of images (probe images) with vision-language model and utilize them as language guidance to train the LG-CAV. Furthermore, after training high-quality LG-CAVs related to all the predicted classes in the target model, we propose the activation sample reweighting (ASR), serving as a model correction technique, to improve the performance of the target model in return. Experiments on four datasets across nine architectures demonstrate that LG-CAV achieves significantly superior quality to previous CAV methods given any concept, and our model correction method achieves state-of-the-art performance compared to existing concept-based methods. Our code is available at `https://github.com/hqhQAQ/LG-CAV`.

## 1 Introduction

Concept activation vector (CAV) [16] interprets the pre-trained black-box classification models (target models) by quantifying the significance of a concept to the model predictions. CAV provides intuitive insights to comprehend the intrinsic behavior of black-box models, elucidating the patterns behind their decision-making processes. Owing to its simplicity and effectiveness, it has been followed by numerous studies [10, 36, 11, 1, 40] and extended to diverse domains, such as recommender system [45], 3D shape generation [8], abusive language detection [25], *etc*.

However, the training of CAV usually necessitates an ample amount of high-quality images that accurately depict the corresponding concept. Unfortunately, in practical contexts, gathering an adequate number of training images is challenging especially when the number of concepts is extensive, thereby significantly impacting the quality (estimated using the proposed *concept accuracy* and *concept-to-class accuracy*) of the trained CAVs. Figure 1 delineates the correlation between

---

† Corresponding author.

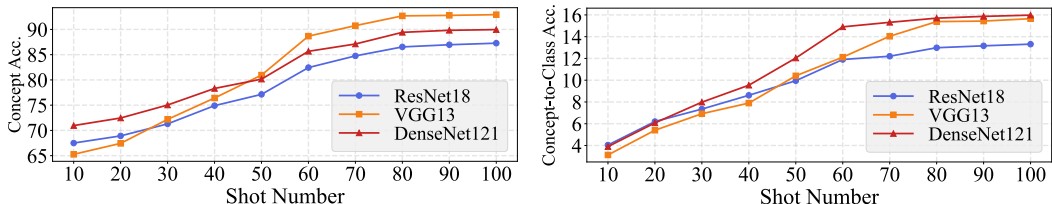

Figure 1: The quality of CAV is significantly affected by the number of training images. Here concept accuracy estimates whether the CAV faithfully represents its corresponding concept. Concept-to-class accuracy measures the similarity between the CAV and its strongly semantic-related class.

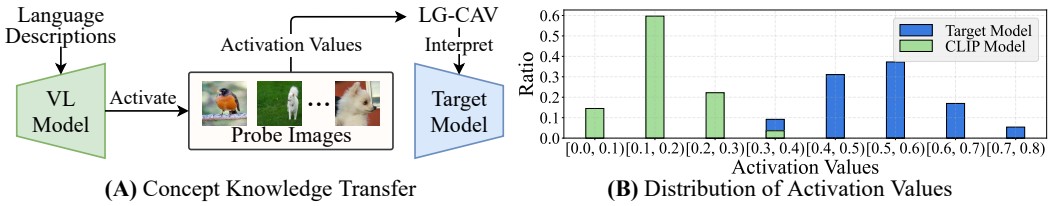

| (A) Concept Knowledge Transfer | (B) Distribution of Activation Values |

Figure 2: **(A)** LG-CAV is trained guided by activations of concept descriptions on the probe images from VL model. **(B)** The distribution of activation values on a concept named "Skyscraper" (from the Broden dataset [2]) in the target model (ResNet18) and VL model (CLIP) differs a lot.

the number of training images for each concept and the quality of the trained CAVs on the Broden dataset [2]. It can be concluded that when the number of training images is small, the quality of the trained CAVs is low, hindering CAVs from properly interpreting the model.

In recent years, the advent of foundational vision-language models (referred to as **VL models**, such as CLIP [30]) establishes connections between images and text by mapping image features and text features into a shared feature space. These VL models undergo pre-training on extensive image-text datasets, equipping them with the ability to grasp a multitude of concepts. Inspired by this, to address the data-scarcity problem of CAV training, in this work we propose LG-CAV to utilize the abundant concept knowledge from VL models for more cheaply getting CAV for any concept given the concept descriptions, without being confined to specific pre-defined concepts.

The concept features extracted by VL model cannot be directly used for training the LG-CAV, as VL model and the target model operate within distinct feature spaces. To bridge the gap, our work ingeniously trains the LG-CAV by calculating its activation values on a common pool of images (probe images), and making them mimic the activation values of the corresponding concept from VL model on these images, as shown in Figure 2 **(A)**. Therefore, LG-CAV learns its corresponding concept according to the concept's existence degree (activation value) on the probe images from VL model.

However, directly applying the above framework is not guaranteed to improve the quality of LG-CAV (see experiments in subsubsection 4.1.2), because the calculated activation values from the target model and VL model are in different distributions (see Figure 2 **(B)**). To tackle this problem, our work proposes a Gaussian alignment (GA) module to align the activation values from the target and VL models. Besides, we propose a concept ensemble (CE) module and a deviation sample reweighting (DSR) module into this framework to further improve the quality of LG-CAV. Detailedly, CE module strengthens the completeness of concept descriptions by employing data augmentations on the concept texts. DSR module optimizes the selection of probe images by allocating higher training weights to the probe images with a more stable concept representation.

Furthermore, after training numerous high-quality LG-CAVs that can describe all classes in the dataset, our work makes a considerable improvement on previous CAV methods by applying LG-CAVs to model correction on generic datasets like ImageNet. To this end, we fine-tune the target model to align the prediction of each class with its strongly-related concept, with a proposed activation sample reweighting (ASR) module that allocates higher training weights to the samples activated more highly by the corresponding LG-CAVs.

We perform extensive experiments to validate the performance of our proposed method. Experiments demonstrate that LG-CAV achieves significantly higher CAV quality (concept accuracy & concept-to-

class accuracy) than previous CAV methods on the Broden & ImageNet datasets over nine backbones. Besides, we conduct model correction on the ImageNet & CUB-200-2011 & CIFAR-100 datasets over nine backbones. Experiments present that our method achieves significantly superior performance to other concept-based methods.

To sum up, the key contributions of this work can be listed as follows:

- We propose LG-CAV to tackle the data-scarcity problem of CAV training, which is trained guided by the corresponding concept descriptions from VL model.

- We propose a Gaussian alignment (GA) module, a concept ensemble (CE) module, and a deviation sample reweighting (DSR) module to further enhance the quality of LG-CAV.

- Beyond providing explanations, we apply LG-CAV to model correction, by proposing an activation sample reweighting (ASR) module.

- Experiment results verify that LG-CAV achieves significantly higher CAV quality, and our model correction method outperforms existing concept-based methods remarkably.

## 2 Related Work

**Concept Activation Vector (CAV).**   With the development and widespread application of deep learning [12, 31, 3, 46], it has become increasingly important to explain the internal mechanisms of deep neural networks (*e.g.*, using concept activation vector (CAV)). Each CAV [16] is trained for a specific concept in the target model, and is used to quantify the importance of this concept to model predictions. Most existing CAV methods only utilize CAVs to interpret the target model. Concept_Gradient [1] extends the original linear CAV to non-linear concept functions, which improves the interpretability of CAV without the linear separability assumption of CAV. OA-TCAV [40] proposes an adversarial training approach to improve the quality of CAV. Differently, our method achieves significantly superior CAV quality to these methods, by transferring the abundant concept knowledge from VL model.

**Vision-Language Models for Interpretability.**   CLIP-Dissect [27] and DISCOVER [28] utilize CLIP model to describe the neurons inside the target model. Label-free CBM [26] and PCBM [47] utilize CLIP model to generate additional concept annotations for concept bottleneck models [17]. These methods are limited to solely interpreting the target model and lacking the ability to improve the model performance using the explanation results.

**Model Correction.**   Model correction methods aim to improve the target model by introducing corrective information into the model. Most existing methods [32, 22, 24, 19, 41] are limited to customized tasks with narrow scope (*e.g.*, debias the color bias of model representations on the ColorMNIST dataset [22]). Some methods [11, 4] improve the accuracy of generic classification models, but they are limited to small-sized datasets. Differently, our method trains high-quality LG-CAVs that can describe all classes in the dataset, thus facilitating the task of model correction on generic datasets like ImageNet.

We provide more detailed comparisons with the related methods in Appendix C.3.

## 3 Method

### 3.1 Preliminaries

The target model is a pre-trained classification model that receives image $x$ as input and outputs $K$ classification logits, with a backbone $f$ and a final layer $h$. Detailedly, $f$ extracts the image features $f(x) \in \mathbb{R}^{D_f}$ of $x$ ($D_f$ is dimension size), and $h$ is a linear layer that projects $f(x)$ into $K$ classification logits. Note $h(f(x)) \in \mathbb{R}^K$, and $h_k(f(x)) \in \mathbb{R}$ is the classification logit for class $k$.

Concept activation vector (CAV) [16] represents a concept for the target model. Specifically, given positive images ($\mathcal{P}_c$) and negative images ($\mathcal{N}_c$) for the concept $c$, a binary linear classifier is trained on internal features $\{f(x) : x \in \mathcal{P}_c\}$ and $\{f(x) : x \in \mathcal{N}_c\}$ to discriminate $c$, with a classification loss $\mathcal{L}_{cls}$. Finally, the CAV $v_c \in \mathbb{R}^{D_f}$ for $c$ is defined as the weight vector for $c$ in the classifier.

VL model [30] consists of an image encoder $g_{\text{img}}$ that projects input image $\boldsymbol{x}$ into image features $g_{\text{img}}(\boldsymbol{x}) \in \mathbb{R}^{D_{\text{VL}}}$, and a text encoder $g_{\text{text}}$ that projects input texts $\boldsymbol{t}$ into text features $g_{\text{text}}(\boldsymbol{t}) \in \mathbb{R}^{D_{\text{VL}}}$. After trained on a large-scale image-text dataset, $g_{\text{img}}(\boldsymbol{x})$ and $g_{\text{text}}(\boldsymbol{t})$ are projected into the same feature space and can be directly compared.

## 3.2 Evaluation of CAV Quality

We propose two metrics (*concept accuracy* and *concept-to-class accuracy*) to evaluate the CAV quality, based on the definition that CAV quantifies the importance of a concept to the class prediction.

**Concept accuracy.** Concept accuracy estimates whether the CAV faithfully represents its corresponding concept. To this end, the accuracy $\text{Acc}(\boldsymbol{v}_c)$ for CAV $\boldsymbol{v}_c$ is calculated as the test accuracy of the binary classification model. Specifically, let $\mathcal{C}$ denote the set of all concepts, concept accuracy $S_{\text{concept}}$ is finally calculated averagely over all concepts (note that $\|\cdot\|$ denotes cardinality of a set):

$$S_{\text{concept}} = \frac{1}{\|\mathcal{C}\|} \sum_{c \in \mathcal{C}} \text{Acc}(\boldsymbol{v}_c). \tag{1}$$

**Concept-to-class accuracy.** The original CAV simply determines whether the trained CAV $\boldsymbol{v}_c$ has a positive relation to a class $k$ in the target model, by simply determining whether the angle between $\boldsymbol{v}_c$ and $\nabla h_k(f(\boldsymbol{x}))$ (the gradients of classification logit for class $k$ on $f(\boldsymbol{x})$) is acute. However, this metric is too simplified to reflect the degree of connection between CAVs and classes. Therefore, we propose concept-to-class accuracy to estimate the extent to which the CAV $\boldsymbol{v}_c$ relates to class $k$ according to the cosine similarity between $\boldsymbol{v}_c$ and $\nabla h_k(f(\boldsymbol{x}))$. We construct the ground-truth set ($\mathcal{D}$) of positively-related concept-class pairs by calculating the similarity between the concepts and the class names with a language model (like all-mpnet-base-v2 [39] as used in CLIP-Dissect [27]) and selecting the concept-class pairs with the similarity exceeding a threshold $\epsilon$. Finally, concept-to-class accuracy $S_{\text{concept\_to\_class}}$ is calculated averagely over all ground-truth concept-class pairs $\mathcal{D}$:

$$S_{\text{concept\_to\_class}} = \frac{1}{\|\mathcal{D}\|} \sum_{(c,k) \in \mathcal{D}} \frac{\boldsymbol{v}_c \cdot \nabla h_k(f(\boldsymbol{x}))}{\|\boldsymbol{v}_c\| \|\nabla h_k(f(\boldsymbol{x}))\|}. \tag{2}$$

## 3.3 LG-CAV

In this section, we first propose a **framework** on how to transfer the concept knowledge from VL model to the LG-CAV, then propose three modules into this framework to further improve the quality of LG-CAV: a **Gaussian alignment (GA) module**, a **concept ensemble (CE) module**, and a **deviation sample reweighting (DSR) module**.

### 3.3.1 Framework

The features of concept descriptions extracted by VL model cannot be directly used to supervise the training of CAVs, because VL model and the target model have different feature spaces. Therefore, we propose an ingenious method that transforms the concept knowledge of VL model into activation values on a common pool of images (also named probe images, denoted as $\mathcal{R}$) and trains the LG-CAV from these activation values, inspired by previous concept-based method [7] that adopts probe images to recognize common units of different models.

Specifically, this method consists of three steps to train LG-CAV $\boldsymbol{v}_c$: **(1)** Calculate the activation values $\{\text{Act}_{\boldsymbol{v}_c}(f(\boldsymbol{x})) : \boldsymbol{x} \in \mathcal{R}\}$ of $\boldsymbol{v}_c$ on the image features $\{f(\boldsymbol{x}) : \boldsymbol{x} \in \mathcal{R}\}$ extracted by the target model. **(2)** Calculate the activation values $\{\text{Act}_{g_{\text{text}}(c)}(g_{\text{img}}(\boldsymbol{x})) : \boldsymbol{x} \in \mathcal{R}\}$ of $g_{\text{text}}(c)$ on $\{g_{\text{img}}(\boldsymbol{x}) : \boldsymbol{x} \in \mathcal{R}\}$ using VL model. **(3)** Train $\boldsymbol{v}_c$ by aligning $\{\text{Act}_{\boldsymbol{v}_c}(f(\boldsymbol{x})) : \boldsymbol{x} \in \mathcal{R}\}$ with $\{\text{Act}_{g_{\text{text}}(c)}(g_{\text{img}}(\boldsymbol{x})) : \boldsymbol{x} \in \mathcal{R}\}$, and the corresponding loss function $\mathcal{L}_{\text{LG-CAV}}$ is shown in Equation 3 (note that $\|\cdot\|_2$ denotes the L2 norm, $f$, $g_{\text{text}}$, $g_{\text{img}}$ are frozen, and only $\boldsymbol{v}_c$ is trainable).

$$\mathcal{L}_{\text{LG-CAV}} = \frac{1}{\|\mathcal{R}\|} \sum_{\boldsymbol{x} \in \mathcal{R}} \left( \text{Act}_{\boldsymbol{v}_c}(f(\boldsymbol{x})) - \text{Act}_{g_{\text{text}}(c)}(g_{\text{img}}(\boldsymbol{x})) \right)^2. \tag{3}$$

We calculate activation value as cosine similarity between two vectors (*e.g.*, $\text{Act}_{\boldsymbol{v}_c}(f(\boldsymbol{x})) = \frac{\boldsymbol{v}_c \cdot f(\boldsymbol{x})}{\|\boldsymbol{v}_c\| \|f(\boldsymbol{x})\|}$), because cosine similarity is invariant to the norms of feature vectors which differ a lot in different models. Therefore, the LG-CAV learns to recognize images with the corresponding

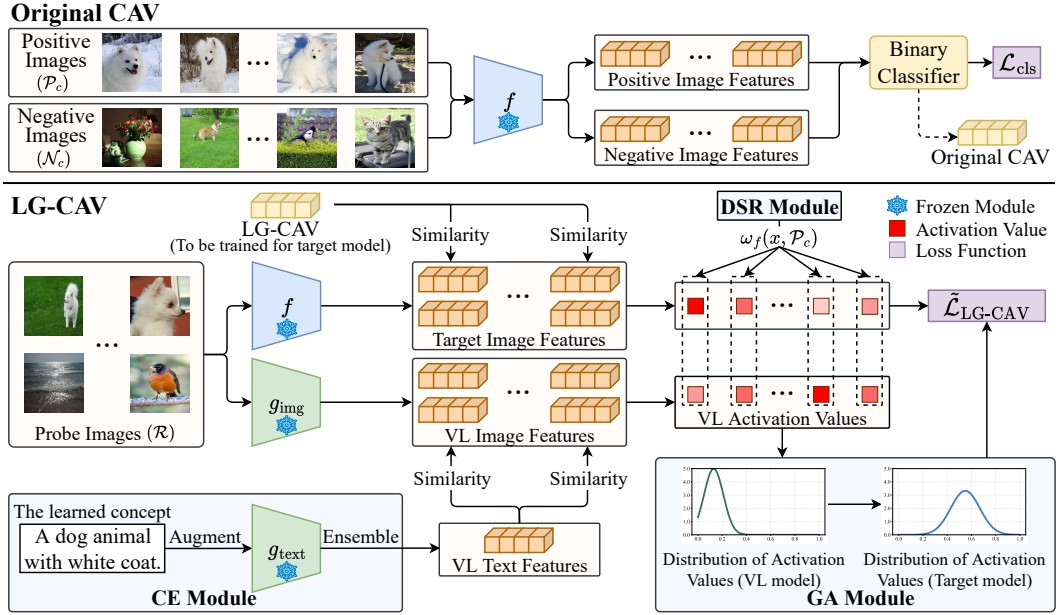

Figure 3: **Top:** The original CAV is defined as the weight vector for its represented concept in the binary linear classifier. **Bottom:** The LG-CAV is learned by mimicking the activation values of its represented concept on the probe images $\mathcal{R}$ using VL model. Besides, three modules (GA module, CE module, and DSR module) are proposed to enhance the quality of LG-CAV.

concept and the images without the corresponding concept. Besides, compared with the original binary classification task for CAV training, the activation values encompass richer information about the extent to which the concepts exist in the images, thus facilitating the training of LG-CAV.

### 3.3.2 Gaussian Alignment Module

However, directly utilizing the above $\mathcal{L}_{\text{LG-CAV}}$ is not guaranteed to improve the quality of CAVs (see experiments in subsubsection 4.1.2), because the activation values calculated from VL model and the target model have significantly different distributions (due to the huge difference of feature space in the two models). To address this problem, Gaussian alignment (GA) module aligns the distribution of activation values for VL model with that for the target model, based on the observation that the distribution of activation values resembles a Gaussian distribution (Figure 2 **(B)**). GA module consists of three steps: **(1)** Calculate the cosine similarity for each pair of features in $\{f(\boldsymbol{x}) : \boldsymbol{x} \in \mathcal{R}\}$ to simulate the activation values from the target model, which will be $\mathcal{A} = \{ \frac{f(\boldsymbol{x}') \cdot f(\boldsymbol{x}'')}{\|f(\boldsymbol{x}')\|\|f(\boldsymbol{x}'')\|} : \boldsymbol{x}', \boldsymbol{x}'' \in \mathcal{R}\}$. **(2)** Estimate the parameters (mean & standard deviation) of Gaussian distribution $X \sim \mathcal{N}(\mu_{\text{target}}, \sigma_{\text{target}}^2)$ for $\mathcal{A}$ (activation values from the target model), and $X \sim \mathcal{N}(\mu_{\text{VL}}, \sigma_{\text{VL}}^2)$ for $\{\text{Act}_{g_{\text{text}}(c)}(g_{\text{img}}(\boldsymbol{x})) : \boldsymbol{x} \in \mathcal{R}\}$ (activation values from VL model). **(3)** Calculate the transformation function for these two Gaussian distributions, then use it to transform each $\text{Act}_{g_{\text{text}}(c)}(g_{\text{img}}(\boldsymbol{x}))$ to be $\tilde{\text{Act}}_{g_{\text{text}}(c)}(g_{\text{img}}(\boldsymbol{x}))$, as shown in Equation 4.

$$\tilde{\text{Act}}_{g_{\text{text}}(c)}(g_{\text{img}}(\boldsymbol{x})) = \frac{\text{Act}_{g_{\text{text}}(c)}(g_{\text{img}}(\boldsymbol{x})) - \mu_{\text{VL}}}{\sigma_{\text{VL}}} \cdot \sigma_{\text{target}} + \mu_{\text{target}}. \tag{4}$$

Detailedly, this transformation first transforms $X \sim \mathcal{N}(\mu_{\text{VL}}, \sigma_{\text{VL}}^2)$ into a standard Gaussian distribution ($X \sim \mathcal{N}(0, 1)$), then transforms the standard Gaussian distribution into $X \sim \mathcal{N}(\mu_{\text{target}}, \sigma_{\text{target}}^2)$, as shown in Appendix A.

### 3.3.3 Concept Ensemble Module

Concept ensemble (CE) module employs data augmentations on the concept descriptions, thus enhancing the comprehensiveness of the concept. Specifically, instead of using a single prompt like "a photo of the concept $c$" (that will be fed into $g_{\text{text}}$), CE module uses multiple prompts (*e.g.*, "a

bright photo of the concept $c$", "a cropped photo of the concept $c$") to describe $c$. These concept prompts follow the class prompts in the original CLIP model, as demonstrated in Appendix C.2. Next, $g_{\text{text}}$ will encode these augmented prompts into text features, and generate the augmented text features $\tilde{g}_{\text{text}}(c)$ by averaging them.

### 3.3.4 Deviation Sample Reweighting Module

Deviation sample reweighting (DSR) module optimizes the selection of probe images, by allocating higher training weights to the probe images that can more stably represent the concept. To this end, DSR module estimates the weight of the probe image $\boldsymbol{x}$ according to the standard deviation of its similarities with the ground-truth positive images $\mathcal{P}_c$, using three steps: **(1)** Calculate $\cos_f(\boldsymbol{x}, \mathcal{P}_c) = \{\frac{f(\boldsymbol{x}) \cdot f(\boldsymbol{x}')}{\|f(\boldsymbol{x})\|\|f(\boldsymbol{x}')\|} : \boldsymbol{x}' \in \mathcal{P}_c\}$. **(2)** Calculate the standard deviation $\text{std}_f(\boldsymbol{x}, \mathcal{P}_c)$ of $\cos_f(\boldsymbol{x}, \mathcal{P}_c)$. Note that $\text{std}_f(\boldsymbol{x}, \mathcal{P}_c) \in \mathbb{R}$, and lower $\text{std}_f(\boldsymbol{x}, \mathcal{P}_c)$ indicates more stable concept representation of $\boldsymbol{x}$. **(3)** The weight $\boldsymbol{\omega}_f(\boldsymbol{x}, \mathcal{P}_c)$ is finally calculated by normalizing the opposite of $\text{std}_f(\boldsymbol{x}, \mathcal{P}_c)$ with a softmax operation, as shown in Equation 5. Note that the averaged value of all sample weights equals 1, and the softmax function can be replaced by other normalization functions.

$$\boldsymbol{\omega}_f(\boldsymbol{x}, \mathcal{P}_c) = \|\mathcal{R}\| \cdot \frac{\exp\left(-\text{std}_f(\boldsymbol{x}, \mathcal{P}_c)\right)}{\sum\limits_{\boldsymbol{x}' \in \mathcal{R}} \exp\left(-\text{std}_f(\boldsymbol{x}', \mathcal{P}_c)\right)}. \tag{5}$$

### 3.3.5 Loss Function

With the above three modules, the updated LG-CAV loss $\tilde{\mathcal{L}}_{\text{LG-CAV}}$ is calculated as in Equation 6. Besides, when positive images $\mathcal{P}_c$ and negative images $\mathcal{N}_c$ are provided, $\tilde{\mathcal{L}}_{\text{LG-CAV}}$ can be added into the training framework of original CAV to enhance the CAV quality, and the total loss function $\mathcal{L}_{\text{total}}$ will be $\mathcal{L}_{\text{total}} = \mathcal{L}_{\text{cls}} + \tilde{\mathcal{L}}_{\text{LG-CAV}}$ (note that $\mathcal{L}_{\text{cls}}$ is the classification loss for the original CAV).

$$\tilde{\mathcal{L}}_{\text{LG-CAV}} = \frac{1}{\|\mathcal{R}\|} \sum_{\boldsymbol{x} \in \mathcal{R}} \boldsymbol{\omega}_f(\boldsymbol{x}, \mathcal{P}_c) \cdot \left(\text{Act}_{\boldsymbol{v}_c}(f(\boldsymbol{x})) - \tilde{\text{Act}}_{\tilde{g}_{\text{text}}(c)}(g_{\text{img}}(\boldsymbol{x}))\right)^2. \tag{6}$$

### 3.4 Model Correction

Due to the lack of high-quality CAVs that can sufficiently relate to all classes in the dataset, most existing CAV methods are confined to explaining the local behavior of target model using a very limited number and variety of CAVs. Different from these methods, our proposed method can train a sufficient quantity of high-quality LG-CAVs that relate to all classes in the dataset, thus having great potential to improve the performance of target model in an interpretable manner.

Specifically, our model correction method alleviates spurious correlation in the target model (*i.e.*, incorrect dependence of a class on unrelated concepts) to improve the model performance. To this end, we fine-tune the target model to align the prediction of each class with its strongly-related LG-CAV. However, directly aligning the gradients for each class with the LG-CAV would easily interfere with other correct concepts and hurt the performance. To align them in a more soft manner, activation sample reweighting (ASR) module allocates different training weights to the images of each class, according to the activation values of the corresponding LG-CAV on them. Assume concept $c$ is strongly related to class $k$, and let $\mathcal{I}_k$ denote the training images of class $k$, then ASR module reweights image $\boldsymbol{x}$ of $\mathcal{I}_k$ in two steps (similar to DSR module): **(1)** Calculate $\text{Act}_{\boldsymbol{v}_c}(f(\boldsymbol{x}))$ (the activation value of LG-CAV $\boldsymbol{v}_c$ on $\boldsymbol{x}$). **(2)** Calculate the weight $\boldsymbol{\omega}_f^{\text{fine-tune}}(\boldsymbol{x})$ by normalizing $\text{Act}_{\boldsymbol{v}_c}(f(\boldsymbol{x}))$ with a softmax operation, as shown in Equation 7.

$$\boldsymbol{\omega}_f^{\text{fine-tune}}(\boldsymbol{x}) = \|\mathcal{I}_k\| \cdot \frac{\exp\left(\text{Act}_{\boldsymbol{v}_c}(f(\boldsymbol{x}))\right)}{\sum\limits_{\boldsymbol{x}' \in \mathcal{I}_k} \exp\left(\text{Act}_{\boldsymbol{v}_c}(f(\boldsymbol{x}'))\right)}. \tag{7}$$

Next, $\boldsymbol{\omega}_f^{\text{fine-tune}}(\boldsymbol{x})$ will be used as the weight of image $\boldsymbol{x}$ in the classification loss during fine-tuning. In this manner, the target model learns to predict class $k$ from the samples activated more highly by the LG-CAV $\boldsymbol{v}_c$, thus better aligning the prediction of class $k$ with its strongly-related concept $c$. Besides, this method requires no further training on the backbone $f$ (only uses $f$ to extract image features and trains the subsequent layers), leading to minimal training cost.

Table 1: The comprehensive evaluation of **concept accuracy** (%) for different CAVs on the Broden dataset. The results are on nine backbones pre-trained on ImageNet (Note that **Res** denotes ResNet, **Dense** denotes DenseNet) averaged over 4 runs with different seeds. Bold font denotes the best result.

| Method | Res-18 | Res-34 | Res-50 | Dense-121 | Dense-169 | VGG-13 | VGG-19 | ViT-B | ViT-L |
|---|---|---|---|---|---|---|---|---|---|
| Original CAV [16] | 68.92 | 69.32 | 71.34 | 72.46 | 72.70 | 67.44 | 69.56 | 65.35 | 65.85 |
| Text-to-Concept [23] | 70.04 | 71.35 | 72.40 | 73.67 | 74.19 | 68.35 | 70.24 | 67.22 | 66.27 |
| OA-TCAV [40] | 72.62 | 72.20 | 73.24 | 73.90 | 74.89 | 68.69 | 70.81 | 67.83 | 67.35 |
| **Ours** | 67.23 | 67.48 | 67.52 | 69.43 | 68.46 | 65.99 | 67.94 | 63.16 | 63.22 |
| **Ours + GA** | 74.89 | 73.47 | 74.19 | 76.28 | 74.70 | 69.63 | 72.31 | 68.99 | 68.41 |
| **Ours + GA + CE** | 76.41 | 74.47 | 75.63 | 78.18 | 76.12 | 70.25 | 72.92 | 69.43 | 69.31 |
| **Ours + GA + CE + DSR** | **77.45** | **76.04** | **76.48** | **79.07** | **77.25** | **70.69** | **73.47** | **70.52** | **70.09** |

Table 2: The comprehensive evaluation of **concept-to-class accuracy** for different CAVs on the Broden dataset averaged over 4 runs with different seeds.

| Method | Res-18 | Res-34 | Res-50 | Dense-121 | Dense-169 | VGG-13 | VGG-19 | ViT-B | ViT-L |
|---|---|---|---|---|---|---|---|---|---|
| Original CAV [16] | 6.20 | 7.02 | 7.20 | 6.08 | 6.54 | 5.40 | 5.53 | 7.22 | 7.97 |
| Text-to-Concept [23] | 9.48 | 9.06 | 8.73 | 7.42 | 8.33 | 7.52 | 7.07 | 10.70 | 11.09 |
| OA-TCAV [40] | 10.11 | 10.29 | 10.90 | 9.18 | 10.63 | 8.38 | 8.74 | 10.07 | 10.68 |
| **Ours** | 4.72 | 6.66 | 5.99 | 5.64 | 6.02 | 4.02 | 4.49 | 6.07 | 6.38 |
| **Ours + GA** | 16.72 | 16.61 | 16.92 | 15.47 | 15.81 | 14.55 | 15.04 | 17.52 | 17.83 |
| **Ours + GA + CE** | 19.14 | 20.10 | 20.51 | 18.78 | 19.00 | 17.50 | 18.35 | 21.52 | 22.34 |
| **Ours + GA + CE + DSR** | **24.58** | **25.61** | **26.05** | **23.93** | **23.97** | **21.40** | **22.79** | **26.12** | **27.72** |

# 4 Experiments

## 4.1 The Quality of LG-CAV

### 4.1.1 Experiment Settings

**Datasets.** We estimate the quality of LG-CAV on the Broden dataset [2] (a popular concept-based dataset with 63,305 images for 1197 visual concepts). In the Broden dataset, each image may contain multiple concepts. Therefore, we collect positive samples for each concept by selecting the images containing only this concept, and randomly select the same number of images from other concepts as negative samples. Finally, the simplified Broden dataset consists of 17,746 images for 468 visual concepts. The probe images ($\mathcal{R}$) for each LG-CAV are from ImageNet and the images of other concepts in the Broden dataset. Specifically, we select the most activated and the same number of most least activated images by VL model.

**Backbones.** We follow the original CAV work [16] to train CAVs for the target models pre-trained on ImageNet (from the open-sourced PyTorch package [29]). These backbones include ResNet [12], DenseNet [14], VGG [38], and Vision Transformer [6].

**Parameters.** To simulate the absence of images for training CAVs in reality, we set the number of positive samples ($\mathcal{P}_c$) and negative samples ($\mathcal{N}_c$) to be 10, and the remaining images will be used as the test set. The threshold $\epsilon$ for determining positively-related concept-class pair is 0.6. For each CAV method, we use SGD optimizer [34] to train the CAV for 10 epochs with a learning rate of 1e-3. $\|\mathcal{R}\|$ (the number of probe images) is set to be 1000. The loss function adopted here is $\mathcal{L}_{\text{total}}$ since $\mathcal{P}_c$ and $\mathcal{N}_c$ are available.

### 4.1.2 Experiment Results

**Concept accuracy.** Table 1 demonstrates that without sufficient data, the accuracy of original CAV is insufficient to accurately represent a concept. The first version of LG-CAV (**Ours**) has a lower accuracy than the original CAV when no other modules are added, due to the large difference in the distribution of activation values. The added GA module aligns the activation values from target model and VL model, and improves the concept accuracy by 5.83 points averagely. The added CE and DSR modules both effectively improve the concept accuracy, and the final LG-CAV outperforms Text-to-Concept and OA-TCAV (see Appendix C.3 for the analysis of them) by a large margin.

**Concept-to-class accuracy.** Table 2 demonstrates that our proposed modules also enhance concept-to-class accuracy, because the CAV that better represents a concept can more accurately correspond

Table 3: The comprehensive evaluation of accuracy (%) on selected classes (40 classes) of ImageNet averaged over 4 runs with different seeds.

| Method | Res-18 | Res-34 | Res-50 | Dense-121 | Dense-169 | VGG-13 | VGG-19 | ViT-B | ViT-L |
|---|---|---|---|---|---|---|---|---|---|
| Original | 90.65 | 91.00 | 92.50 | 92.00 | 92.75 | 88.55 | 90.90 | 94.55 | 93.15 |
| HiBug [4] | 90.38 | 90.82 | 92.69 | 92.36 | 92.60 | 88.74 | 91.07 | 94.74 | 93.46 |
| **Ours** | **91.16** | **91.79** | **93.06** | **92.91** | **93.16** | **89.21** | **91.43** | **94.94** | **93.66** |

Table 4: The comprehensive evaluation of accuracy (%) for different methods on ImageNet (note that KD denotes knowledge distillation) averaged over 4 runs with different seeds.

| Method | Res-18 | Res-34 | Res-50 | Dense-121 | Dense-169 | VGG-13 | VGG-19 | ViT-B | ViT-L |
|---|---|---|---|---|---|---|---|---|---|
| Original | 69.76 | 73.31 | 76.13 | 74.43 | 75.60 | 69.93 | 72.38 | 81.07 | 79.67 |
| Concept_Distillation [11] | 69.46 | 73.06 | 75.77 | 74.04 | 75.46 | 69.80 | 72.29 | 80.86 | 79.51 |
| KD [13] | 69.93 | 73.49 | 76.27 | 74.68 | 75.99 | 70.06 | 72.46 | 81.15 | 79.77 |
| Label-free CBM [26] | N/A | N/A | 71.95 | N/A | N/A | N/A | N/A | N/A | N/A |
| **Ours** | **70.26** | **73.66** | **76.47** | **74.94** | **76.28** | **70.19** | **72.60** | **81.38** | **80.05** |

to its strongly-related class. The final LG-CAV also has much higher concept-to-class accuracy than Text-to-Concept and OA-TCAV.

## 4.2 Model Correction

### 4.2.1 Experiment Settings

**Datasets.** We employ our model correction method on three representative datasets: ImageNet [5] (large-scale dataset), CUB-200-2011 [43] (a popular dataset used by many concept-based methods), and CIFAR-100 [18] (small-scale dataset). The probe images ($\mathcal{R}$) for each LG-CAV are also the most highly and least activated images by VL model from their respective datasets.

**Backbones.** The target models pre-trained on ImageNet are from PyTorch, and the target models pre-trained on CIFAR-100 and CUB-200-2011 are from another open-sourced PyTorchCV package, following PCBM [47]. The VL model adopted here is CLIP model with ViT-L/14 as backbone.

**Parameters.** We use SGD optimizer to train the final classification layer for 20 epochs with a learning rate of 1e-3. Note that different from the experiments in subsection 4.1, the training of LG-CAVs for these concepts does not require the original classification loss $\mathcal{L}_{\text{cls}}$ and DSR module, due to the lack of ground-truth positive samples $\mathcal{P}_c$.

### 4.2.2 Experiment Results

We adopt two methods to find the strongly-related concept of each class in the target model, corresponding to two types of datasets: **datasets with few classes** & **datasets with many classes**.

**Datasets with few classes.** We manually collect the concept descriptions of each class from Wikipedia for these datasets (*e.g.*, the randomly selected subset of ImageNet with 40 classes (ImageNet-40)). The selected classes and their corresponding concept descriptions can be referred to in Appendix C.1.

Appendix B.1 demonstrates that the trained LG-CAVs have ability to distinguish whether images contain their respective concepts. Next, we utilize these LG-CAVs for model correction with the ASR module. As shown in Table 3, our model correction method effectively improves the performance of original pre-trained model (converted from the pre-trained 1000-classes model by removing other 960 classes in the final classification layer), by an improvement of up to 0.91 points.

**Datasets with many classes.** Collecting sufficient high-quality concept descriptions for datasets with many classes is a challenging task. Therefore, we instead acquire the concept descriptions of each class based on its comparison with its confused class, inspired by relative CAV proposed in the original CAV work. Specifically, for the class $k$, we first find the confused class $k'$ to which images from class $k$ are most likely to be mispredicted by the pre-trained model, then define the concept descriptions as "a photo of class $k$, not $k'$". This approach is applied to the whole ImageNet, CUB-200-2011, and CIFAR-100.

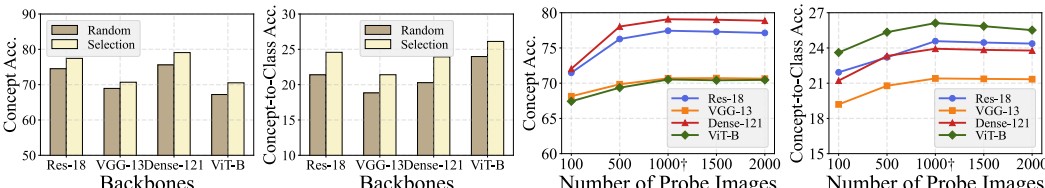

Figure 4: Ablation experiments on probe images (selection strategy & image number).

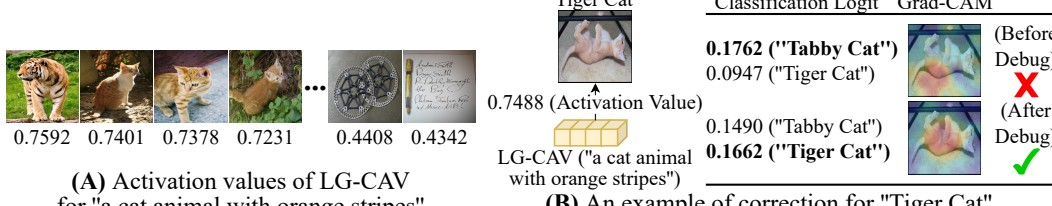

**(A)** Activation values of LG-CAV
for "a cat animal with orange stripes"

**(B)** An example of correction for "Tiger Cat"

Figure 5: **(A)** Activation values of the LG-CAV & **(B)** Model correction example.

As shown in Table 4, our method improves model performance on the whole ImageNet in an interpretable manner based on LG-CAV, surpassing original model (by up to 0.68 points), Concept_Distillation, knowledge distillation, and Label-free CBM (see Appendix C.3 for the analysis of them). Besides, the results on CUB-200-2011 (Appendix B.2) & CIFAR-100 (Appendix B.3) also verify the effectiveness of our method.

## 4.3 Ablation Study

**Selection of probe images.** In the above experiments, we select the most activated and least activated images as probe images $\mathcal{R}$. We compare this selection strategy with random selection in this subsection. As shown in the first two figures of Figure 4, the LG-CAVs trained with this selection strategy have higher quality, because the probe images selected by this strategy contain richer information to represent the corresponding concepts.

**Number of probe images.** This subsection investigates how the number of probe images affects the quality of LG-CAV. As shown in the last two figures of Figure 4, when the number of probe images is small, increasing the quantity of probe images can improve the quality of LG-CAV. However, when the number of probe images reaches a certain level (saturation), further increasing the quantity of probe images does not improve the quality of LG-CAV.

Additionally, we provide more ablation experiments on the choice of VL model, the coefficient of LG-CAV loss, and the depth of extracted image features in the target model in Appendix B.5.

## 4.4 Visualization Results

**Activation values of the trained LG-CAV.** Figure 5 **(A)** demonstrates the activation values of a trained LG-CAV on its highly-activated and lowly-activated images, indicating that the trained LG-CAV can accurately activate images that contain the corresponding concepts.

**Examples of model correction.** As shown in Figure 5 **(B)**, an image of "Tiger Cat" is misclassified as "Tabby Cat" by the target model (with ResNet18 as backbone) before model correction. During model correction, ASR module mitigates spurious correlation of the target model by aligning the prediction of "Tiger Cat" with its strongly-related concept "a cat animal with orange stripes". This image is activated by the LG-CAV of this concept with a high activation value (0.7488), thus the classification logit for "Tiger Cat" increases after model correction. Besides, we utilize Grad-CAM [37] to attribute the prediction of "Tiger Cat" in the target model, and it shows that the attribution map focuses more accurately on the cat's body after model correction.

Furthermore, we provide more visualization results in Appendix D.

## 5 Conclusion

In this work, we propose LG-CAV to address the data-scarcity problem of original CAV by transferring the extensive concept knowledge from VL model. Specifically, LG-CAV mimics the activation values from VL model on the probe images to learn these concept knowledge. Besides, we propose a Gaussian alignment (GA) module, a concept ensemble (CE) module, and a deviation sample reweighting (DSR) module to further enhance the quality of LG-CAV. Furthermore, we go beyond previous CAV methods by generalizing LG-CAV to model correction, with a human-understandable method that aligns the class predictions with the strongly-related concepts. Experiment results demonstrate that LG-CAV significantly improves the CAV quality, and our model correction method outperforms existing concept-based methods by a large margin. We hope our work can provide inspiration for future interpretable methods based on vision-language models.

## Acknowledgements

This work is supported by National Natural Science Foundation of China (62106220, U20B2066), Alibaba-Zhejiang University Joint Research Institute of Frontier Technologies, and Zhejiang Province "Pioneering Soldier" and "Leading Goose" R&D Project under Grant 2023C01027.

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

# A Proof

To address the problem that activation values calculated from VL model and the target model have significantly different distributions (due to the huge difference of feature space in the two models), our proposed Gaussian alignment (GA) module first estimates the Gaussian distribution of activation values for the target model and VL model ($X_{\mathrm{VL}} \sim \mathcal{N}(\mu_{\mathrm{VL}}, \sigma^2_{\mathrm{VL}})$ and $X_{\mathrm{target}} \sim \mathcal{N}(\mu_{\mathrm{target}}, \sigma^2_{\mathrm{target}})$), then transforms each $\mathrm{Act}_{g_{\mathrm{text}}(c)}(g_{\mathrm{img}}(\boldsymbol{x}))$ to be $\tilde{\mathrm{Act}}_{g_{\mathrm{text}}(c)}(g_{\mathrm{img}}(\boldsymbol{x}))$ according to the transformation between these two Gaussian distributions:

$$\tilde{\mathrm{Act}}_{g_{\mathrm{text}}(c)}(g_{\mathrm{img}}(\boldsymbol{x})) = \frac{\mathrm{Act}_{g_{\mathrm{text}}(c)}(g_{\mathrm{img}}(\boldsymbol{x})) - \mu_{\mathrm{VL}}}{\sigma_{\mathrm{VL}}} \cdot \sigma_{\mathrm{target}} + \mu_{\mathrm{target}}. \tag{8}$$

Specifically, this transformation first transforms $X \sim \mathcal{N}(\mu_{\mathrm{VL}}, \sigma^2_{\mathrm{VL}})$ into a standard Gaussian distribution ($X \sim \mathcal{N}(0, 1)$), then transforms the standard Gaussian distribution into $X \sim \mathcal{N}(\mu_{\mathrm{target}}, \sigma^2_{\mathrm{target}})$, which can be proved in the following theorem.

**Theorem:** The Gaussian distribution $X_{\mathrm{VL}} \sim \mathcal{N}(\mu_{\mathrm{VL}}, \sigma^2_{\mathrm{VL}})$ of activation values for VL model can be converted into Gaussian distribution $X_{\mathrm{target}} \sim \mathcal{N}(\mu_{\mathrm{target}}, \sigma^2_{\mathrm{target}})$ of activation values for the target model with a linear transformation: $X_{\mathrm{target}} = \frac{X_{\mathrm{VL}} - \mu_{\mathrm{VL}}}{\sigma_{\mathrm{VL}}} \cdot \sigma_{\mathrm{target}} + \mu_{\mathrm{target}}$.

*Proof.* We first prove that $X_{\mathrm{VL}} \sim \mathcal{N}(\mu_{\mathrm{VL}}, \sigma^2_{\mathrm{VL}})$ can be converted into the standard Gaussian distribution $X_{\mathrm{standard}} \sim \mathcal{N}(0, 1)$ with a linear transformation: $\frac{X_{\mathrm{VL}} - \mu_{\mathrm{VL}}}{\sigma_{\mathrm{VL}}}$.

As Gaussian distributions, the cumulative distribution function of $X_{\mathrm{VL}}$ is $F_{X_{\mathrm{VL}}}(k) = \mathrm{P}(X_{\mathrm{VL}} \leq k) = \int_{-\infty}^{k} \frac{1}{\sqrt{2\pi}\sigma_{\mathrm{VL}}} \exp\left(-\frac{(x-\mu_{\mathrm{VL}})^2}{2\sigma^2_{\mathrm{VL}}}\right) \mathrm{d}x$, and the cumulative distribution function of $X_{\mathrm{standard}}$ is $F_{X_{\mathrm{standard}}}(k) = \mathrm{P}(X_{\mathrm{standard}} \leq k) = \int_{-\infty}^{k} \frac{1}{\sqrt{2\pi}} \exp\left(-\frac{x^2}{2}\right) \mathrm{d}x$.

Let $X = \frac{X_{\mathrm{VL}} - \mu_{\mathrm{VL}}}{\sigma_{\mathrm{VL}}}$, then the cumulative distribution function of $X$ will be $F_X(k) = \mathrm{P}(X \leq k) = \mathrm{P}(\frac{X_{\mathrm{VL}} - \mu_{\mathrm{VL}}}{\sigma_{\mathrm{VL}}} \leq k) = \mathrm{P}(X_{\mathrm{VL}} \leq k \cdot \sigma_{\mathrm{VL}} + \mu_{\mathrm{VL}}) = \int_{-\infty}^{k \cdot \sigma_{\mathrm{VL}} + \mu_{\mathrm{VL}}} \frac{1}{\sqrt{2\pi}\sigma_{\mathrm{VL}}} \exp\left(-\frac{(x-\mu_{\mathrm{VL}})^2}{2\sigma^2_{\mathrm{VL}}}\right) \mathrm{d}x$. Next, let $x = z \cdot \sigma_{\mathrm{VL}} + \mu_{\mathrm{VL}}$, then $\mathrm{d}x = \sigma_{\mathrm{VL}}\mathrm{d}z$, and $F_X(k) = \int_{-\infty}^{k} \frac{1}{\sqrt{2\pi}\sigma_{\mathrm{VL}}} \exp\left(-\frac{(z \cdot \sigma_{\mathrm{VL}})^2}{2\sigma^2_{\mathrm{VL}}}\right)\sigma_{\mathrm{VL}}\mathrm{d}z = \int_{-\infty}^{k} \frac{1}{\sqrt{2\pi}} \exp\left(-\frac{z^2}{2}\right)\mathrm{d}z$. Therefore, $F_X(k)$ (the cumulative distribution function of $X$) is equal to $F_{X_{\mathrm{standard}}}(k)$ (the cumulative distribution function of $X_{\mathrm{standard}}$), proving that $X$ is identical with $X_{\mathrm{standard}}$.

Likewise, $X_{\mathrm{target}} \sim \mathcal{N}(\mu_{\mathrm{target}}, \sigma^2_{\mathrm{target}})$ can be converted into the standard Gaussian distribution $X_{\mathrm{standard}} \sim \mathcal{N}(0, 1)$ with a linear transformation: $\frac{X_{\mathrm{target}} - \mu_{\mathrm{target}}}{\sigma_{\mathrm{target}}}$.

$$\begin{cases} X_{\mathrm{standard}} = \frac{X_{\mathrm{VL}} - \mu_{\mathrm{VL}}}{\sigma_{\mathrm{VL}}}. \\ X_{\mathrm{standard}} = \frac{X_{\mathrm{target}} - \mu_{\mathrm{target}}}{\sigma_{\mathrm{target}}}. \end{cases} \tag{9}$$

Therefore, by combining these two equations, $X_{\mathrm{target}}$ can be converted from $X_{\mathrm{VL}}$ with a linear transformation, as shown in Equation 10.

$$\begin{aligned} X_{\mathrm{target}} &= X_{\mathrm{standard}} \cdot \sigma_{\mathrm{target}} + \mu_{\mathrm{target}} \\ &= \frac{X_{\mathrm{VL}} - \mu_{\mathrm{VL}}}{\sigma_{\mathrm{VL}}} \cdot \sigma_{\mathrm{target}} + \mu_{\mathrm{target}}. \end{aligned} \tag{10}$$

$\square$

# B Experiments

## B.1 LG-CAV Quality

For the datasets with few classes, we manually collect the strongly-related concept descriptions of each class from Wikipedia. After training the LG-CAVs for these concept descriptions, we first verify

Table 5: The averaged Recall@100 (%) of LG-CAVs on ImageNet-40. Random CAV denotes the randomly initialized CAV. Bold font denotes the best result.

| Method | Res-18 | Res-34 | Res-50 | Dense-121 | Dense-169 | VGG-13 | VGG-19 | ViT-B | ViT-L |
|---|---|---|---|---|---|---|---|---|---|
| Random CAV | 4.98 | 5.07 | 4.62 | 5.00 | 5.04 | 4.60 | 4.74 | 4.92 | 5.08 |
| **LG-CAV** | **76.25** | **73.13** | **76.88** | **75.63** | **79.38** | **83.75** | **85.63** | **84.38** | **78.75** |

Table 6: The evaluation of accuracy (%) for different concept-based methods on the CUB-200-2011 dataset over five backbones. The results of PCBM & Trustworthy CBM & Label-free CBM are from their original paper.

| Method | Res-10 | Res-12 | Res-14 | Res-16 | Res-18 |
|---|---|---|---|---|---|
| Original | 72.23 | 72.73 | 75.23 | 76.35 | 76.67 |
| PCBM [47] | N/A | N/A | N/A | N/A | 58.80 |
| PCBM-h [47] | N/A | N/A | N/A | N/A | 61.00 |
| Trustworthy CBM [15] | N/A | N/A | N/A | N/A | 68.30 |
| Label-free CBM [26] | N/A | N/A | N/A | N/A | 74.31 |
| **Ours** | **72.74** | **73.14** | **75.66** | **76.67** | **77.31** |

the quality of LG-CAVs by calculating the Recall@100 performance of them. Specifically, given an LG-CAV, we use CLIP model (with ViT-L/14 as backbone) to find 50 test images that best match its corresponding concept as ground-truth, then calculate the Recall@100 by comparing them with the 100 most activated test images calculated from LG-CAV. Table 5 illustrates the averaged Recall@100 performance over all LG-CAVs, indicating that the trained LG-CAVs have the ability to distinguish images containing their respective concepts.

## B.2  Model Correction on CUB-200-2011

Table 6 demonstrates the experiment results of model correction on CUB-200-2011 over five backbones (ResNet10, ResNet12, ResNet14, ResNet16, and ResNet20), indicating that our model correction method shows superior performance to the original model. Besides, our method naturally exceeds other concept-based interpretability methods (PCBM [47], Trustworthy CBM [15], and Label-free CBM [26]) that sacrifice performance for the sake of interpretability.

## B.3  Model Correction on CIFAR-100

Table 7 also verifies the effectiveness of our method on small-scale dataset (CIFAR-100) over five backbones (ResNet20, DenseNet40, PreResNet20, SEResNet20, and SEPreResNet20).

Table 7: The evaluation of accuracy (%) for different methods on the CIFAR-100 dataset over five backbones.

| Method | Res-20 | Dense-40 | Pre-Res-20 | SE-Res-20 | SE-Pre-Res-20 |
|---|---|---|---|---|---|
| Original | 70.36 | 75.10 | 69.78 | 71.46 | 71.69 |
| **Ours** | **70.87** | **75.59** | **70.19** | **71.87** | **71.94** |

## B.4  TCAV Score

In the main paper, we use concept-to-class accuracy to estimate the similarity between the CAV and its strongly semantic-related class. The original CAV adopts a simpler but incomplete metric named TCAV score for this purpose, which estimates whether the trained CAV $v_c$ has a positive relation to its positively-related class $k$ by simply determining whether the angle between $v_c$ and $\nabla h_k(f(x))$ (the gradients of classification logit for class $k$) is acute. Table 8 demonstrates that our proposed modules also effectively improve the TCAV score.

Table 8: The comprehensive evaluation of **TCAV score** (%) for different CAVs on the Broden dataset. Bold font denotes the best result.

| Method | Res-18 | Res-34 | Res-50 | Dense-121 | Dense-169 | VGG-13 | VGG-19 | ViT-B | ViT-L |
|---|---|---|---|---|---|---|---|---|---|
| Original CAV | 76.15 | 82.31 | 83.85 | 83.08 | 90.00 | 88.46 | 86.15 | 81.54 | 83.85 |
| **Ours + GA + CE + DSR** | **96.15** | **98.46** | **98.24** | **98.28** | **99.03** | **99.23** | **97.69** | **94.62** | **93.85** |

## B.5 Additional Ablation Study

### B.5.1 Different VL Models

We conduct ablation experiments on how the choice of CLIP model affects the quality of LG-CAV. As shown in Table 9 and Table 10, the LG-CAVs trained from CLIP models with ViTs (ViT-L/14, ViT-B/16, and ViT-B/32) as backbone have higher quality than those trained from CLIP models with CNNs (RN50×16) as backbone, because the former CLIP models have much higher performance (zero-shot accuracy) than the latter ones. Besides, Table 9 and Table 10 demonstrates that the quality of LG-CAVs increases in the order of ViT-L/14 $\to$ ViT-B/16 $\to$ ViT-B/32, indicating that the ViTs with larger patch sizes lead to LG-CAVs with higher quality. This is because ViTs with larger patch size focus more on the overall concepts in the images rather than specific local details, making the learned concept features easier to transfer to the target model.

Table 9: The comprehensive evaluation of concept accuracy (%) with different CLIP models in four target backbones (ResNet18, DenseNet121, VGG13, and ViT-B/16) on Broden.

| Method | RN50×16 | ViT-L/14 | ViT-B/16 | ViT-B/32 |
|---|---|---|---|---|
| Res-18 | 76.12 | 77.45 | 77.85 | 78.22 |
| Dense-121 | 77.89 | 79.07 | 79.46 | 79.59 |
| VGG-13 | 70.02 | 70.69 | 71.11 | 71.39 |
| ViT-B | 70.15 | 70.52 | 70.88 | 70.99 |

Table 10: The comprehensive evaluation of concept-to-class accuracy with different CLIP models in four target backbones (ResNet18, DenseNet121, VGG13, and ViT-B/16) on Broden.

| Method | RN50×16 | ViT-L/14 | ViT-B/16 | ViT-B/32 |
|---|---|---|---|---|
| Res-18 | 24.13 | 24.58 | 25.98 | 26.23 |
| Dense-121 | 23.65 | 23.93 | 25.23 | 25.56 |
| VGG-13 | 21.52 | 21.40 | 22.33 | 22.78 |
| ViT-B | 25.92 | 26.12 | 26.26 | 26.94 |

Besides, we adopt other VL models (EVA-CLIP [42], LaCLIP [9], CLIPA [21]) to train the LG-CAVs. These VL models are advanced variants of the original CLIP model with more accurate vision-language alignment, and the LG-CAVs trained with these VL models have higher concept accuracy and concept-to-class accuracy, as shown in Table 11 and Table 12.

### B.5.2 Coefficient of LG-CAV Loss

Figure 6 demonstrates how the coefficient of LG-CAV loss affects the quality of LG-CAV. Initially, increasing the loss coefficient will increase the quality of LG-CAV (in both concept accuracy and concept-to-class accuracy). However, when it exceeds 3.0, further increasing it will decrease the quality of LG-CAV.

### B.5.3 CAV Quality on Intermediate Features

In the experiments, we utilize the image features extracted from the last layer of the backbone to train CAVs, because deep features better capture high-level concepts. Besides, we also conduct experiments on the intermediate features of the target model over three backbones (ResNet18, DenseNet121, and ViT-B). Specifically, the depth of layer for extracting the intermediate features of these backbones is 13, 88, 11, respectively.

Table 11: The comprehensive evaluation of concept accuracy (%) with different VL models in four target backbones (ResNet18, DenseNet121, VGG13, and ViT-B/16) on Broden. These VL models all adopt ViT-L/14 as backbones.

| Method | Res-18 | Dense-121 | VGG-13 | ViT-B |
|---|---|---|---|---|
| Original CAV | 68.92 | 72.46 | 67.44 | 65.35 |
| LG-CAV (CLIP) | 77.45 | 79.07 | 70.69 | 70.52 |
| LG-CAV (EVA-CLIP) | 79.90 | 82.13 | 73.53 | 71.69 |
| LG-CAV (LaCLIP) | 77.93 | 79.95 | 71.39 | 71.06 |
| LG-CAV (CLIPA) | 78.45 | 80.16 | 71.42 | 70.80 |

Table 12: The comprehensive evaluation of concept-to-class accuracy (%) with different VL models in four target backbones (ResNet18, DenseNet121, VGG13, and ViT-B/16) on Broden. These VL models all adopt ViT-L/14 as backbones.

| Method | Res-18 | Dense-121 | VGG-13 | ViT-B |
|---|---|---|---|---|
| Original CAV | 6.20 | 6.08 | 5.40 | 7.22 |
| LG-CAV (CLIP) | 24.58 | 23.93 | 21.40 | 26.12 |
| LG-CAV (EVA-CLIP) | 25.60 | 24.79 | 22.20 | 28.56 |
| LG-CAV (LaCLIP) | 26.29 | 25.68 | 22.69 | 28.44 |
| LG-CAV (CLIPA) | 25.46 | 24.85 | 22.10 | 28.23 |

As shown in Table 13 and Table 14, our proposed modules can still effectively improve the quality of LG-CAV (in both concept accuracy and concept-to-class accuracy) on the intermediate features. However, compared with the features extracted from the last layer, the LG-CAVs trained from intermediate features have much lower quality. Besides, intermediate features have a much larger dimension size than the features extracted from the last layer, resulting in enormous training costs. Therefore, these two factors hinder the CAVs trained from intermediate features for broader applications.

Table 13: The comprehensive evaluation of concept accuracy (%) for intermediate features in three backbones (ResNet18, DenseNet121, and ViT-B/16) of the target model on the Broden dataset. Bold font denotes the best result.

| Method | Res-18 | Dense-121 | ViT-B |
|---|---|---|---|
| Original CAV | 64.68 | 66.96 | 56.04 |
| **Ours** | 57.36 | 56.46 | 54.96 |
| **Ours + GA** | 68.21 | 70.65 | 58.56 |
| **Ours + GA + CE** | 69.25 | 71.09 | 59.45 |
| **Ours + GA + CE + DSR** | **70.99** | **72.31** | **60.84** |

## B.6 Standard Deviation

The experiment results of our main experiments are averaged over 4 runs with different seeds, but the standard deviations of experiment results are omitted in the main paper due to space limit. Table 15, Table 16, and Table 17 demonstrate the standard deviation of experiment results, indicating that the results are relatively stable in the experiments of **evaluation of CAV quality** and **model correction**. The results of four representative backbones from ResNet, DenseNet, VGG, and ViT are demonstrated, and the results of other backbones are similar.

## C   More Experiment Details

### C.1   Concept Descriptions of 40 Classes in ImageNet

For the datasets with few classes (*e.g.*, the randomly selected subset of ImageNet with 40 classes (ImageNet-40)), we manually collect the concept descriptions of each class from Wikipedia for these datasets, as shown in Table 18.

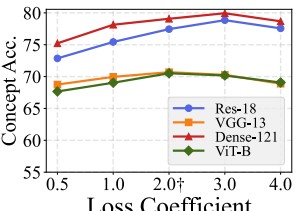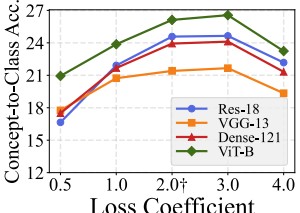

Figure 6: Ablation experiments on coefficient of LG-CAV loss. † denotes the loss coefficient used in the experiments of the main paper.

Table 14: The comprehensive evaluation of concept-to-class accuracy for intermediate features in three backbones (ResNet18, DenseNet121, and ViT-B/16) on Broden. Bold font denotes the best result.

| Method | Res-18 | Dense-121 | ViT-B |
|---|---|---|---|
| Original CAV | 0.29 | 0.29 | 0.56 |
| **Ours** | 0.20 | 0.23 | 0.48 |
| **Ours + GA** | 0.58 | 0.63 | 0.68 |
| **Ours + GA + CE** | 0.86 | 1.04 | 0.85 |
| **Ours + GA + CE + DSR** | **1.40** | **1.80** | **1.16** |

## C.2 Data Augmentation Templates in the CE Module

Concept ensemble (CE) module employs data augmentations on the concept descriptions. Specifically, instead of using a single prompt like "a photo of the concept $c$" (that will be fed into $g_{\text{text}}$), CE module uses multiple prompts (*e.g.*, "a bright photo of the concept $c$", "a cropped photo of the concept $c$") to describe $c$. The data augmentation templates are demonstrated in Table 19.

## C.3 Comparisons with Baselines

### C.3.1 The Quality of CAVs

We compare LG-CAV with four CAV methods: original CAV [16], Concept Gradient [1], OA-TCAV [40], and Text-to-Concept [23].

**Original CAV [16].** The original CAV is defined as the weight vector for the corresponding concept in the binary linear classifier that classifies the positive images and negative images for this concept. The original CAV has poor quality when the training images for the concept are insufficient.

**Concept Gradient [1].** Concept Gradient extends the original linear CAV to non-linear concept functions, improving the quality of CAV trained on the features extracted from intermediate layers of the target model. In particular, when the features are extracted from the final layer of target model (linearly separable), Concept Gradient is identical to the original CAV.

**OA-TCAV [40].** OA-TCAV proposes an adversarial training approach to improve the quality of CAV. However, it still suffers from the data-scarcity problem, and thus is inferior to our method.

Table 15: STD of **concept accuracy** (%) for different CAVs on the Broden dataset. The results are on nine backbones pre-trained on ImageNet.

| Method | Res-18 | Dense-121 | VGG-13 | ViT-B |
|---|---|---|---|---|
| Original CAV [16] | $68.92 \pm 0.35$ | $72.46 \pm 0.54$ | $67.44 \pm 0.34$ | $65.35 \pm 0.41$ |
| Text-to-Concept [23] | $70.04 \pm 0.71$ | $73.67 \pm 0.50$ | $68.35 \pm 0.76$ | $67.22 \pm 0.55$ |
| OA-TCAV [40] | $72.62 \pm 0.18$ | $73.90 \pm 0.40$ | $68.69 \pm 0.30$ | $67.83 \pm 0.32$ |
| **Ours** | $67.23 \pm 0.41$ | $69.43 \pm 0.48$ | $65.99 \pm 0.46$ | $63.16 \pm 0.46$ |
| **Ours + GA** | $74.89 \pm 0.57$ | $76.28 \pm 0.26$ | $69.63 \pm 0.49$ | $68.99 \pm 0.46$ |
| **Ours + GA + CE** | $76.41 \pm 0.44$ | $78.18 \pm 0.41$ | $70.25 \pm 0.61$ | $69.43 \pm 0.53$ |
| **Ours + GA + CE + DSR** | $\mathbf{77.25} \pm 0.38$ | $\mathbf{79.07} \pm 0.31$ | $\mathbf{70.69} \pm 0.50$ | $\mathbf{70.52} \pm 0.23$ |

Table 16: STD of **concept-to-class accuracy** for different CAVs on the Broden dataset averaged over 4 runs with different seeds.

| Method | Res-18 | Dense-121 | VGG-13 | ViT-B |
|---|---|---|---|---|
| Original CAV [16] | $6.20 \pm 0.49$ | $6.08 \pm 0.55$ | $5.40 \pm 0.47$ | $7.22 \pm 0.66$ |
| Text-to-Concept [23] | $9.48 \pm 0.97$ | $7.42 \pm 1.05$ | $7.52 \pm 1.13$ | $10.70 \pm 1.24$ |
| OA-TCAV [40] | $10.11 \pm 0.38$ | $9.18 \pm 0.43$ | $8.38 \pm 0.40$ | $10.07 \pm 0.32$ |
| **Ours** | $4.72 \pm 1.19$ | $5.64 \pm 0.77$ | $4.02 \pm 1.10$ | $6.07 \pm 1.13$ |
| **Ours + GA** | $16.72 \pm 0.79$ | $15.47 \pm 1.17$ | $14.55 \pm 1.08$ | $17.52 \pm 0.94$ |
| **Ours + GA + CE** | $19.14 \pm 0.94$ | $18.78 \pm 0.76$ | $17.50 \pm 0.74$ | $21.52 \pm 1.15$ |
| **Ours + GA + CE + DSR** | $\mathbf{24.58} \pm 1.16$ | $\mathbf{23.93} \pm 0.90$ | $\mathbf{21.40} \pm 1.25$ | $\mathbf{26.12} \pm 0.91$ |

Table 17: STD of accuracy (%) for different methods on ImageNet averaged over 4 runs with different seeds.

| Method | Res-18 | Dense-121 | VGG-13 | ViT-B |
|---|---|---|---|---|
| Original | 69.76 | 74.43 | 69.93 | 81.07 |
| Concept_Distillation | $69.46 \pm 0.06$ | $74.04 \pm 0.06$ | $69.80 \pm 0.05$ | $80.86 \pm 0.03$ |
| KD | $69.93 \pm 0.05$ | $74.68 \pm 0.04$ | $70.06 \pm 0.04$ | $81.15 \pm 0.03$ |
| **Ours** | $\mathbf{70.26} \pm 0.04$ | $\mathbf{74.94} \pm 0.02$ | $\mathbf{70.19} \pm 0.03$ | $\mathbf{81.38} \pm 0.02$ |

**Text-to-Concept [23].** Similar to our proposed LG-CAV, Text-to-Concept leverages VL model to generate CAVs, by directly mapping the features of VL model into the feature space of target model. However, it roughly conducts feature mapping with a linear projection matrix, without specialized optimization for each individual CAV like our LG-CAV. Therefore, the quality of CAV trained with Text-to-Concept is inferior to LG-CAV. Besides, Text-to-Concept can only be applied to small-sized datasets with few classes (*e.g.*, IN9 dataset [44], Living17 dataset [35]), without generalization ability to large datasets like ImageNet.

### C.3.2 Model Correction

We compare our model correction method with four baselines: Concept_Distillation [11], Knowledge Distillation [13], Label-free CBM [26], and HiBug [4].

**Concept_Distillation [11].** Concept_Distillation mitigates spurious correlation in the target model by directly aligning the gradients for each class with the LG-CAV using cosine similarity. However, this approach is not applicable to generic datasets like ImageNet, because it would easily interfere with other correct concepts for each class and hurt the performance, as shown in Table 17.

**Knowledge Distillation [13].** Knowledge distillation transfers the knowledge from VL model to the target model by transferring the probabilistic predictions from VL model. For comparison with our method, we freeze the backbone of target model and only train the final classification layer using knowledge distillation.

**Label-free CBM [26].** Label-free CBM incorporates an intermediate concept layer into the target model, and makes class predictions based on the prior concept predictions. As a concept-based interpretable model, Label-free CBM sacrifices more performance to achieve higher model transparency, and thus naturally performs worse than our method.

**HiBug [4].** HiBug leverages pre-trained large language models like ChatGPT and pre-trained vision-language models like BLIP [20] to interpret the target model, and repair the model by training it on the generated data from stable diffusion model [33]. HiBug is limited to small-sized datasets (ImageNet-40) because the data generation cost is too large for large datasets.

## D  More Visualization Results

We provide more visualization results generated by our method. Specifically, Figure 7 demonstrates the highly activated images (and the activation values) of LG-CAVs for eight concepts on the ResNet18 backbone. Figure 8 demonstrates eight model correction examples for the target model on the ResNet18 backbone.

| Class Name | Concept Descriptions |
|---|---|
| Electric Ray | A round marine animal with a stocky tail |
| African Rock Python | A python animal with a small triangular head |
| Yellow Garden Spider | A spider animal with red or yellow portions near the body |
| Partridge | A bird animal with thick neck and rounded wings |
| Toy Terrier | A dog animal with white coat and a short, high-set tail |
| Black and Tan Coonhound | A dog animal with long ears and a strong tail |
| English Foxhound | A dog animal with thick skull and long muzzle |
| Otterhound | A dog animal with dense shaggy coat and webbed feet |
| Norfolk Terrier | A dog animal with hard, wiry, and straight coat |
| Wire Fox Terrier | A dog animal with wiry and dense double coat, triangular head, dark eyes, and small V-shaped ears |
| Golden Retriever | A dog animal with long tails and dark eyes |
| Australian Kelpie | A dog animal with a lean, muscular build and soft short coat |
| Siberian Husky | A dog animal with thick, double-coated fur, pointed ears and bushy tail |
| Toy Poodle | A dog animal with distinctive dense and curly coat |
| Red Fox | A fox animal with a long, bushy tail, a narrow, pointed muzzle, and thick, soft fur |
| Tiger Cat | A cat animal with orange, gold, and red stripes |
| Leaf Beetle | An insect animal with solid and tough body |
| Gazelle | A gazelle animal with tan buff coats and white rumps |
| Bath Towel | A rectangular, thin object made of fabric |
| Bathtub | A long, usually rectangular container |
| Cassette | A flat, rectangular container made of plastic |
| Candy Store | A room with an assortment of sweets |
| Desktop Computer | A computer with a rectangular chassis |
| Doormat | A rectangular piece of fabric material |
| Gong | A flat, circular metal disc |
| Hair Spray | A pressurized aerosol can |
| Hatchet | A small, handheld tool with sharp blade |
| Hook | A curved or bent piece made of metal or plastic |
| Laptop Computer | A portable computer with a rectangular display screen |
| Tights | A garment similar to leggings but is thinner |
| Overskirt | A short skirt |
| Product Packet / Packaging | A container that holds the product |
| Paddle | A relatively flat object with a long handle |
| Soup Bowl | A small, round container for serving soups |
| Electrical Switch | A rectangular or square shape, with a small lever or button |
| Toilet Seat | A flat or curved seating surface on top of a toilet bowl |
| Velvet Fabric | A soft fabric with smooth and lustrous surface |
| Wall Clock | A typically circular-shaped clock face with numbers |
| Eggnog | A creamy beverage with pale yellow or off-white color |
| Cliff | A vertical or near-vertical rock exposure |

Table 18: Concept descriptions of 40 classes from ImageNet.

| Data Augmentation Templates | |
| --- | --- |
| a photo of the {}. | a low resolution photo of the {}. |
| a rendering of the {}. | graffiti of the {}. |
| a bad photo of the {}. | a cropped photo of the {}. |
| a bright photo of the {}. | a drawing of the {}. |
| a photo of the cool {}. | a close-up photo of the {}. |
| a painting of the {}. | a pixelated photo of the {}. |
| a sculpture of the {}. | a plastic {}. |
| a photo of the dirty {}. | a jpeg corrupted photo of the {}. |
| a blurry photo of the {}. | a photo of the hard to see {}. |
| a good photo of the {}. | a close-up photo of the {}. |
| the origami {}. | a sketch of the {}. |
| a photo of the clean {}. | a photo of the large {}. |
| a photo of the nice {}. | a photo of the weird {}. |
| a photo of the small {}. | a black and white photo of the {}. |
| a dark photo of the {}. | |

Table 19: The data augmentation templates for the concept descriptions in the CE module.

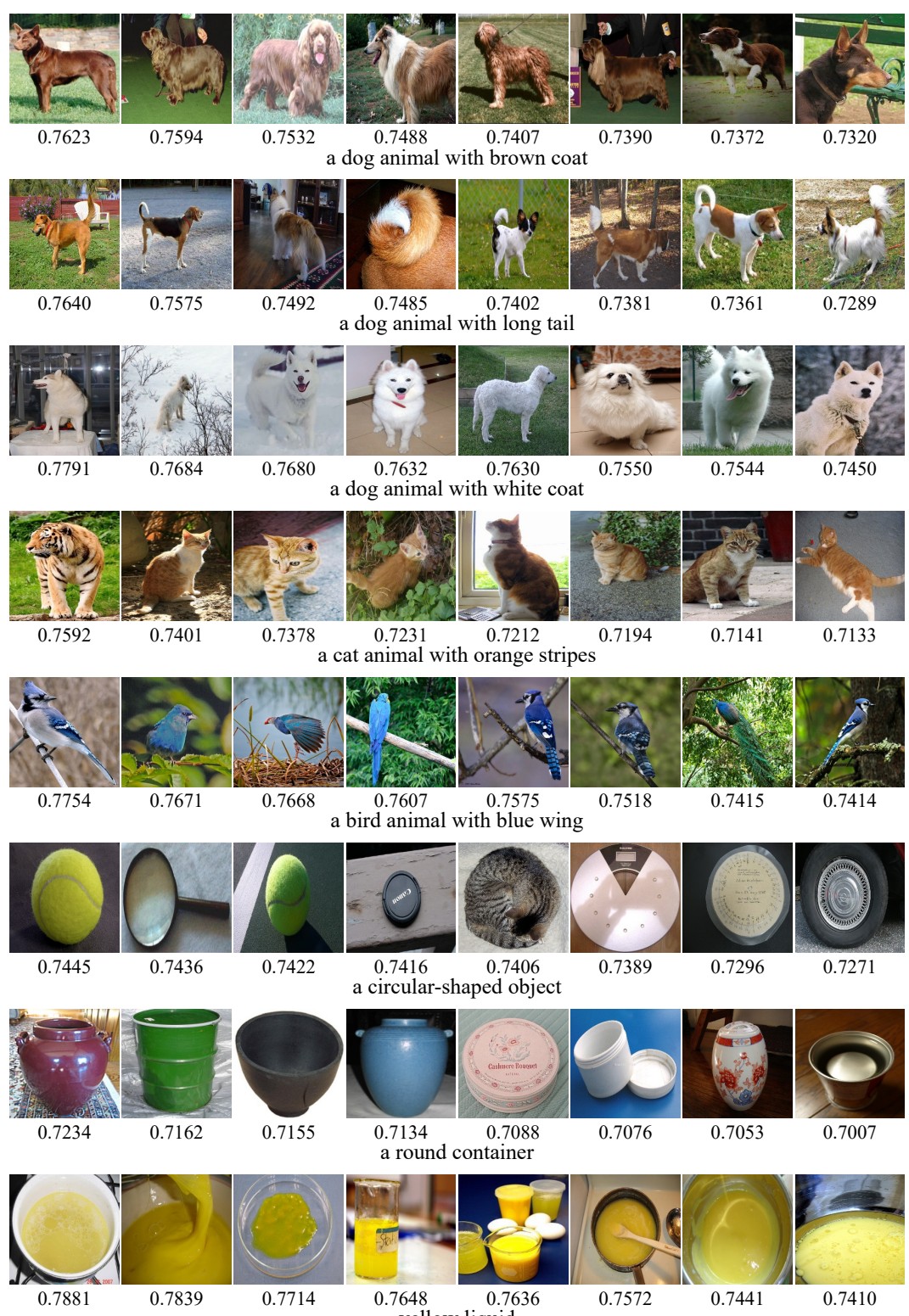

Figure 7: Highly activated images (and the activation values) of LG-CAVs.

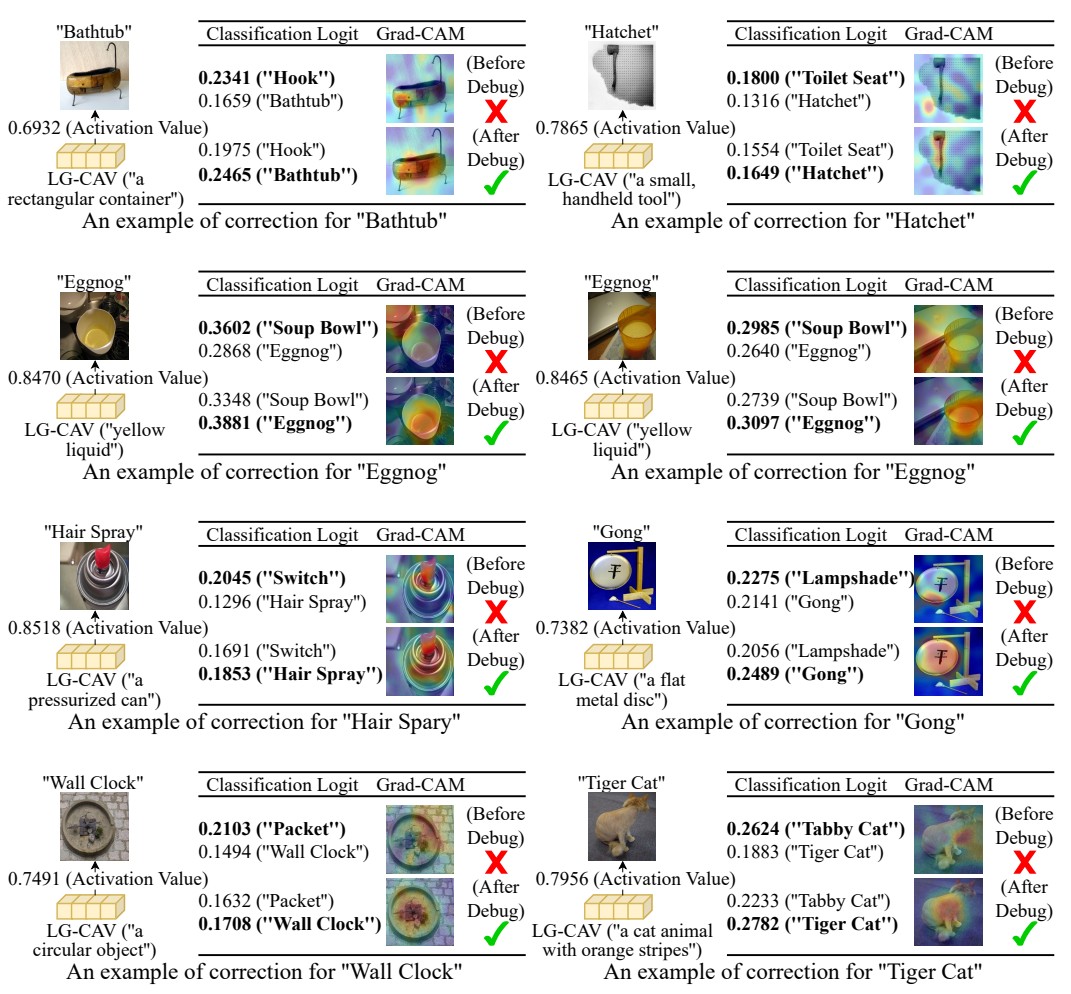

Figure 8: Model correction examples.

