# OpenReview forum: "LG-CAV: Train Any Concept Activation Vector with Language Guidance"
_NeurIPS.cc/2024/Conference — NeurIPS 2024 poster_

### Official Review · Reviewer_sN5p · 2024-07-02

**Soundness:** 3
**Presentation:** 3
**Contribution:** 3
**Rating:** 6
**Confidence:** 4

**Summary:**

This paper proposed a LG-CAV model that leverage the pretrained vision language model to train CAV without label. This includes a concept ensemble model that employ data augmentation on concept text, a DSR module that optmize the selection of probe image and a model to align prediction of class to concepts called ASR. Experiments showed that this framework achives higher CAV quality.

**Strengths:**

The methodology is carried out clearly. The problem is important and the authors have got some good results. The experiments show its proposed model’s effectiveness.

**Weaknesses:**

1.Since LG-CAV leverage pretrained model, I am wondering whether this framework can handle unseen class besides just supervised setting such as novel category detection/generalized category detection. For example, instead of pick 40 classes from ImageNet, could you use 20 for training and the other 20 for testing?

2. From figure 5 we can see that not all concepts are related to the input sentence. I am curious how the similarity threshold is selected.

3. Evaluation of concept-to-class accuracy will need human evaluation of concept. How this is done in detail?

4. From Table 4, the improvement compare with LG-CAV and other baseline is limited.

**Questions:**

see the weaknesses

---

> ### Author Rebuttal · Authors · 2024-08-06
>
> Thank you for dedicating your time and effort to providing valuable suggestions for this paper! We will rigorously revise the paper based on your review!
>
> > **Weakness 1.** Since LG-CAV leverage pretrained model, I am wondering whether this framework can handle unseen class besides just supervised setting such as novel category detection/generalized category detection. For example, instead of pick 40 classes from ImageNet, could you use 20 for training and the other 20 for testing?
>
> **Response:** The LG-CAVs trained on one dataset can be transferred to another dataset, by utilizing the knowledge from the pre-trained VL model. We verify this on a more challenging setting by transferring the LG-CAVs trained on ImageNet to the Stanford Dogs dataset and the CUB-200-2011 dataset, instead of splitting the 40 classes into 20 training classes and 20 test classes (since the used backbones pre-trained on ImageNet have access to the total 40 classes).
>
> Specifically, we train the LG-CAVs for each class of Stanford Dogs and CUB-200-2011, using ImageNet data as probe images. Next, we use each trained LG-CAV as the weight of a binary classifier, and test its classification accuracy on the test images from the corresponding class and the same number of test images from other classes. Table 1 and Table 2 demonstrate the average classification accuracy of the LG-CAVs trained using only ImgeNet data can achieve 80% to 90%, without a large performance gap with the LG-CAVs trained on the original datasets.
>
> Table 1: LG-CAV accuracy on Stanford Dogs with different training data.
> |Training Data|ResNet18|DenseNet121|ViT-B|
> |:-----|:-----|:-----|:-----|
> |ImageNet|84.76|87.68|93.73|
> |Stanfords Dog|86.97|94.02|97.05|
>
> Table 2: LG-CAV accuracy on CUB-200-2011 with different training data.
> |Training Data|ResNet18|DenseNet121|ViT-B|
> |:-----|:-----|:-----|:-----|
> |ImageNet|77.53|80.41|85.53|
> |CUB-200-2011|80.86|84.52|87.60|
>
> > **Weakness 2.** From figure 5 we can see that not all concepts are related to the input sentence. I am curious how the similarity threshold is selected.
>
> **Response:** Actually, Figure 5 demonstrates the high activation values of a trained LG-CAV on the left images and the low activation values on the right images, which indicates that the trained LG-CAV can discriminate whether the images are related to the target concept. We will add more annotation text to this figure for clarification in the revised paper.
>
> > **Weakness 3.** Evaluation of concept-to-class accuracy will need human evaluation of concept. How this is done in detail?
>
> **Response:** Actually, the evaluation of concept-to-class accuracy requires no human evaluation. Instead, we follow the previous work CLIP-Dissect [1] to utilize a pre-trained language model to determine the similarity between a concept and a class, and select the concept-class pairs with high similarities as ground-truth. Next, for each selected concept-class pair, the concept-to-class accuracy of the corresponding CAV is evaluated according to the similarity between the CAV and the corresponding class in the target model.
>
> > **Weakness 4.** From Table 4, the improvement compare with LG-CAV and other baseline is limited.
>
> **Response:** Thanks for pointing out this issue. In Table 4, our model correction method freezes the backbone of the model and only trains the final classification layer, verifying that LG-CAV can mitigate the spurious correlation problem for performance improvement with minimal training cost. This paves the way for training LG-CAVs for the middle layers of the backbone, and using them to supervise the training of backbone for more performance improvement in the future.
>
> [1] Oikarinen et al., Clip-dissect: Automatic description of neuron representations in deep vision networks, ICLR 2023.

---

### Official Review · Reviewer_TRvX · 2024-07-08

**Soundness:** 3
**Presentation:** 4
**Contribution:** 3
**Rating:** 7
**Confidence:** 5

**Summary:**

This paper proposes LG-CAV, a method to train Concept Activation Vectors (CAVs) for any concept without labeled image data, leveraging knowledge from pre-trained vision-language models like CLIP.

The authors introduce several techniques to improve CAV quality, including Gaussian alignment, concept ensemble, and deviation sample reweighting.

They also propose using the trained LG-CAVs for model correction to improve classification performance.

Experiments demonstrate superior CAV quality and model correction results compared to existing methods across multiple datasets and model architectures.

**Strengths:**

1. This paper is well written. The framework is integral, and every component is clearly described in detail. The whole pipeline is easy to follow.

2. The authors propose a novel method with feature alignment loss functions and neural modules to bridge the gap between pre-trained vision-language models and the target classification model.

3. The experimental results show the proposed method achieves the best performance.

4. This paper introduces two new metrics (concept accuracy and concept-to-class accuracy) to evaluate CAV quality. These two metrics are reasonable.

**Weaknesses:**

1. Some of the proposed modules (e.g., Gaussian alignment) seem heuristic. It would be better to give some theoretical justifications.

2. The method uses a set of probe images, but it's not clear how to ensure that the trained LG-CAVs generalize beyond these images.

**Questions:**

No specific questions. This work is of high completeness.

**Limitations:**

No specific limtations. This work is of high completeness.

---

> ### Author Rebuttal · Authors · 2024-08-06
>
> Thank you for investing your time and effort in offering valuable suggestions for this paper! We will rigorously revise the paper based on your review!
>
> > **Weakness 1.** Some of the proposed modules (e.g., Gaussian alignment) seem heuristic. It would be better to give some theoretical justifications.
>
> **Response:** Thanks for the advice! The theoretical analysis on our proposed modules are as follows:
>
> * Gaussian alignment module.
>
>   We first prove here that the mathematic expectation of the deviation between two trained LG-CAVs is positively correlated with the deviation between the distribution of activation values for their training (with Gaussian distribution as an example), indicating that misaligned activation distributions seriously disrupt the LG-CAV training.
>
>   Furthermore, we have provided proof of how GA module aligns the distribution of activation values from VL model to the target model, in Section A of the Appendix.
>
>   **Definition:** Suppose ${\rm Act}_1^{\rm gt},{\rm Act}_2^{\rm gt},...,{\rm Act}_M^{\rm gt}$ are the ground-truth activation values of $M$ probe images for the target model, which follow the Gaussian distribution $\mathcal{N}(\mu _ {\rm gt}, \sigma _ {\rm gt}^2)$. $v_c\in\mathbb{R}^{\rm dim}$ is the LG-CAV, ${\rm Act}_i^{v_c}$ is the activation value of $v_c$ on the $i$-th probe image, and the loss function is $\mathcal{L} _ {\rm gt}=\frac{1}{M}\sum _ {i=1}^{M}({\rm Act}_i^{v_c}-{\rm Act}_i^{\rm gt})^2$. Besides, the deviated activation values ${\rm Act}_1^{\rm shift},{\rm Act}_2^{\rm shift},...,{\rm Act}_M^{\rm shift}$ and loss $\mathcal{L} _ {\rm shift}$ are defined likewise. $v_c^{\rm gt}$ and $v_c^{\rm shift}$ are the LG-CAVs trained with these two losses, respectively. $v[k]$ denotes the $k$-th element of a vector $v$.
>
>   **Theorem:** For each element $k$, the mathematic expectation of $(v_c^{\rm shift}-v_c^{\rm gt})[k]$ is positively correlated with $\mu_{\rm shift}-\mu _ {\rm gt}$.
>
>   *Proof:*
>
>   $$
>   \frac{\partial\mathcal{L} _ {\rm gt}}{\partial\mathcal{v_c}}=\frac{\partial\frac{1}{M}\sum_{i=1}^{M}({\rm Act}_i^{v_c}-{\rm Act}_i^{\rm gt})^2}{\partial\mathcal{v_c}} = \frac{2}{M}\sum _{i=1}^{M}({\rm Act}_i^{v_c}-{\rm Act}_i^{\rm gt})\cdot\frac{\partial{\rm Act}_i^{v_c}}{\partial v_c}.
>   $$
>
>   Gaussian distribution has such properties:
>   * For each $x \sim \mathcal{N}(\mu, \sigma^2)$ and two constants $a$ and $b$, $a x + b \sim \mathcal{N}(a \mu + b, a^2 \sigma^2)$.
>   * For each $x_1 \sim \mathcal{N}(\mu_1, \sigma_1^2)$ and $x_2 \sim \mathcal{N}(\mu_2, \sigma_2^2)$, $x_1 + x_2 \sim \mathcal{N}(\mu_1 + \mu_2, \sigma_1^2 + \sigma_2^2)$.
>
>   Next, at each step of gradient descent, $v_c^{\rm gt}$ is updated as: $v_c^{\rm gt} = v_c - \gamma \frac{\partial \mathcal{L}_{\rm gt}}{\partial \mathcal{v_c}}$, and $v_c^{\rm shift}$ is updated likewise. Therefore, substitute the above formulas into $(v_c^{\rm shift} - v_c^{\rm gt})[k]$, we have:
>
>   $$
>   (v_c^{\rm shift} - v_c^{\rm gt})[k] = \frac{2 \gamma}{M} \sum_{i=1}^{M} ({\rm Act}_i^{\rm shift}-{\rm Act}_i^{\rm gt}) \cdot \frac{\partial {\rm Act}_i^{v_c}}{\partial v_c}[k]
>   $$
>
>   $$
>   \sim \mathcal{N}(\frac{2 \gamma}{M} \cdot \sum _ {i=1}^{M} \frac{\partial {\rm Act}_i^{v_c}}{\partial v_c}[k] \cdot (\mu _ {\rm shift}-\mu _ {\rm gt}), \frac{4 \gamma^2}{M^2} \cdot \sum _ {i=1}^{M} (\frac{\partial {\rm Act}_i^{v_c}}{\partial v_c}[k])^2 \cdot (\sigma _ {\rm shift}^2+\sigma _ {\rm gt}^2)) =  \mathcal{N}(A \cdot (\mu _{\rm shift}-\mu _{\rm gt}), B \cdot (\sigma _{\rm shift}^2+\sigma _{\rm gt}^2)).
>   $$
>
>   Note that $A$ and $B$ are unrelated to the activation distributions, thereby $(v_c^{\rm shift}-v_c^{\rm gt})[k]$ follows a Gaussian distribution, and its mathematic expectation is positively correlated with $\mu_{\rm shift}-\mu_{\rm gt}$. (Some steps are too simplified due to space limit, we will add the full proof into the revised paper.)
>
> * Activation sample reweighting module.
>
>   ASR module allocates higher training weights to the samples with higher activation values on the corresponding LG-CAV. We prove that with this strategy, the class weight in the trained linear classifier will have higher similarity with its corresponding LG-CAV, thus mitigating the spurious correlation problem.
>
>   **Definition:** $\mathcal{I} _ k = \\{ {x_i} \in \mathbb{R}^{\rm dim} \\}_{i=1}^{N}$ denotes the image features of all $N$ training images of class $k$. $u_k \in \mathbb{R}^{\rm dim}$ denotes the class weight for class $k$ in the linear classifier (total $K$ classes in this classifier), and $z _ {i, k} = \langle x_i, u_k \rangle$ denotes the inner product. $\omega_i$ is the weight calculated from ASR module ($\omega_i > 0$), higher $\omega_i$ indicates that the corresponding LG-CAV is more similar with $x_i$.
>
>   **Theorem:** The $u_k$ trained with $\mathcal{L} = -\frac{1}{N} \sum_{i=1}^{N} \omega_i \cdot \log \frac{\exp(z_{i, k})}{\sum_{t=1}^{K} \exp(z_{i,t})}$ (weighted cross-entropy loss) is more similar with the corresponding LG-CAV than the $u_k$ trained with $\mathcal{L} = -\frac{1}{N} \sum_{i=1}^{N} \log \frac{\exp(z_{i, k})}{\sum_{t=1}^{K} \exp(z_{i,t})}$ (original cross-entropy loss).
>
>   We leave out the proof here due to space limit, and will add the proof into the revised paper.
>
> > **Weakness 2.** The method uses a set of probe images, but it's not clear how to ensure that the trained LG-CAVs generalize beyond these images.
>
> **Response:** Our experiments demonstrate that the probe images with more diverse relation (activation values) with the target concept lead to higher performance of the trained LG-CAVs, because these probe images contain richer knowledge for discriminating the target concept. Therefore, in this work we choose to expand the range of probe images (using ImageNet) to increase their diversity and guarantee the generalization ability. Actually, the range of probe images can be easily enlarged because the unlabeled images can be easily obtained from the Internet.

---

### Official Review · Reviewer_V8uU · 2024-07-10

**Soundness:** 2
**Presentation:** 3
**Contribution:** 2
**Rating:** 5
**Confidence:** 3

**Summary:**

The paper introduces Language-Guided Concept Activation Vectors (LG-CAV), a method to train Concept Activation Vectors (CAVs) without labeled data by leveraging pre-trained vision-language models such as CLIP. LG-CAV uses concept descriptions to guide the training of CAVs by aligning the activation values of concept descriptions on a set of probe images. To improve the quality of LG-CAVs, the authors propose three modules: Gaussian Alignment (GA), Concept Ensemble (CE), and Deviation Sample Reweighting (DSR). The paper also introduces an Activation Sample Reweighting (ASR) technique for model correction, which enhances the performance of the target model. Experiments across various datasets and architectures demonstrate that LG-CAV outperforms existing CAV methods in terms of concept accuracy and concept-to-class accuracy.

**Strengths:**

1. The paper is well-written.
2. The use of vision-language models allows for training CAVs without the need for labeled data.
3. The introduction of GA, CE, and DSR modules improves the quality of LG-CAVs.
4. Beyond generating explanations, the method is applied to model correction, leading to improved performance in target models.
5. The results show substantial improvements in both concept accuracy and concept-to-class accuracy compared to existing methods.

**Weaknesses:**

1. The method proposed in this paper does not clearly address the data scarcity problem of the original CAV methods, which was highlighted at the beginning. Although the method is effective, it is not evident why it successfully addresses the data scarcity issue.
2. The method heavily relies on the availability and performance of pre-trained vision-language models like CLIP, which may not always be accessible or optimal for all tasks.
3. The introduction of multiple enhancement modules increases the complexity and computational requirements of the method, which may be a barrier to practical applications.
4. The paper could benefit from a more detailed analysis of scenarios where LG-CAV does not perform well or fails to improve over traditional methods.

**Questions:**

1. Equation 3 uses cosine similarity to calculate the activation values because cosine similarity is invariant to the norms of feature vectors as claimed by the authors. However, this reasoning does not convincingly explain why cosine similarity is necessary in this context.
2. Additionally, in lines 153-155, the authors state that "compared with the original binary classification task for CAV training, the activation values encompass richer information about the extent to which the concepts exist in the images, thus facilitating the training of LG-CAV." I question why richer information can be obtained here. Is there any empirical evidence to support this claim?
3. How does the performance of LG-CAV vary with different types and sizes of pre-trained vision-language models beyond CLIP?
4. What are the computational costs and training times associated with the LG-CAV method compared to traditional CAV methods?

**Limitations:**

See above.

---

> ### Author Rebuttal · Authors · 2024-08-06
>
> Thank you for dedicating your time and effort to provide valuable suggestions for this paper! We will make rigorous revisions to the paper based on the review!
>
> > **Weakness 1. Why it successfully addresses the data scarcity issue.**
>
> **Response:** Previous methods can only train the CAV for the target concept using the **human-collected** positive & negative images. Our method tackles the data scarcity issue by transferring the abundant concept knowledge from CLIP model to LG-CAV for the target model. Specifically,  the LG-CAV is trained by learning the similarity scores of CLIP model on a pool of **unlabeled images** (which can be easily obtained from the Internet).
>
> For example, for a concept named ``a cat animal with orange stripes'', CLIP model can calculate the similarity score of this concept on the unlabeled images, by comparing the text features and the image features. The images that are closer to this concept will be assigned a higher similarity score, owing to the excellent cross-modal ability of CLIP model. After learning these similarity scores, the LG-CAV also has the capacity to discern the similarity between an image and this concept, as shown in Figure 5 (A) of the main paper.
>
> > **Weakness 2. Relies on CLIP model.**
>
> **Response:** CLIP model has demonstrated its excellent cross-modal ability in numerous tasks. By transferring the ability of CLIP model to the target model, our experiments have verified that our method can be applied to universal concepts, e.g., the concepts from the ImageNet dataset and the Broden dataset. Although current CLIP models may have some limitations, the experiments in Section B.5.1 of the Appendix show that our method can be generalized to different types of CLIP models, indicating that it has great potential to be adapted with the next-generation CLIP models in the future.
>
> > **Weakness 3. Computation cost.**
>
> **Response:** Actually, our method only uses an alignment loss function to learn the activation values from VL model, without adding new parameters. Besides, the coefficients of the added modules (*e.g.*, the sample weights of ASR module) can be calculated **only once** before training, requiring no redundant calculation at each step of training and saving the training cost. As shown in Table 1, compared with the original CAV, our method increases only 6% to 7% training time but significantly improves the CAV quality.
>
> Table 1: Training time for 468 concepts in the Broden dataset on one A800 GPU.
> ||ResNet18|DenseNet121|ViT-B|
> |:-----|:-----|:-----|:-----|
> |Original CAV|6.89 Hours|8.68 Hours|8.22 Hours|
> |LG-CAV|7.38 Hours (+7.11%)|9.22 Hours (+6.22%)|8.72 Hours (+6.08%)|
>
> > **Weakness 4. Failure cases.**
>
> **Response:** Our work may fail in some special datasets like MNIST, as CLIP model achieves only 88% zero-shot image classification accuracy on MNIST. Nevertheless, our method can be applied to universal images, and has the ability to be adapted to the next-generation CLIP models with better performance.
>
> > **Question 1. Why cosine similarity is necessary.**
>
> **Response:** Due to the difference in feature dimensions between the target model and VL model, other metrics (e.g., Euclidean distance) can cause significant variation in the similarity scores between features from the target model and CLIP model, making them unable to compare and resulting in poor performance. Cosine similarity naturally mitigates this problem with a normalization operation to constrain the similarity scores from different models within [-1, 1], making them comparable.
>
> Besides, the existing CLIP models also use cosine similarity to compute the similarity between text and image features. We follow this metric to maintain the performance of the CLIP model. As shown in Table 2, cosine similarity achieves better performance than the other two metrics on three backbones.
>
> Table 2: Concept accuracy & concept-to-class accuracy in the Broden dataset with different metrics.
> ||ResNet18|DenseNet121|ViT-B|
> |:-----|:-----|:-----|:-----|
> |Cosine Similarity|77.45 & 24.58|79.07 & 23.93|70.52 & 26.12|
> |Euclidean Distance|56.99 & 4.29|57.42 & 5.24|60.38 & 4.86|
> |Pearson Correlation|76.25 & 21.22|77.87 & 22.79|69.77 & 24.29|
>
> > **Question 2. Why richer information can be obtained here.**
>
> **Response:** This method shares a similar theory with Knowledge Distillation (KD). In KD, the student model learns the classification logits (soft label) of an input image from the teacher model instead of the original hard label, because soft label encompasses richer information about the similarity of the image with every class. Likewise, the activation value in our method indicates the similarity of the image with the target concept, with richer information than only assigning a positive or negative label to the image in traditional CAV.
>
> To verify this, we conduct an experiment by setting the activation values above a high threshold (0.8) to 1, and other activation values are set to -1. As shown in Table 3, the loss of the similarity information leads to worse performance.
>
> Table 3: Concept accuracy & concept-to-class accuracy in the Broden dataset with/without similarity information (hard label).
> ||ResNet18|DenseNet121|ViT-B|
> |:-----|:-----|:-----|:-----|
> |Original|77.45 & 24.58|79.07 & 23.93|70.52 & 26.12|
> |Hard Label|75.28 & 20.69|76.40 & 17.04|68.10 & 16.66|
>
> > **Question 3. Different CLIP models.**
>
> **Response:** We have provided the performance of our method with different VL models in Section B.5.1 of the Appendix. Overall, the VL models with higher zero-shot image classification accuracy lead to higher performance on LG-CAV.
>
> > **Question 4. Computation cost.**
>
> **Response:** The training time compared between our method and the original CAV is shown in Table 1.

---

> > ### Comment · Reviewer_V8uU · 2024-08-11
> >
> > Thank you for your detailed response. After carefully reviewing your clarifications, I believe my initial concern has been addressed. As a result, I will increase my score.

---

### Decision · Program_Chairs · 2024-09-25

**Decision:**

Accept (poster)

**Comment:**

The paper initially received mixed reviews. Reviewer V8uU expressed concerns, giving a negative score, about the rationale for the model’s success, its reliance on a pretrained model, and its increased complexity. However, the author rebuttal successfully addressed these concerns, prompting the reviewer to raise their score. Consequently, all the reviewers acknowledged the value of the work based on its novelty, presentation, and effectiveness. Based on these recommendations, the AC also recommends the paper for acceptance.